# Genomic Diversity in Sporadic Breast Cancer in a Latin American Population

**DOI:** 10.3390/genes11111272

**Published:** 2020-10-28

**Authors:** Lucía Brignoni, Mónica Cappetta, Valentina Colistro, Mónica Sans, Nora Artagaveytia, Carolina Bonilla, Bernardo Bertoni

**Affiliations:** 1Departamento de Genética, Facultad de Medicina, Universidad de la República, Montevideo 11800, Uruguay; lbrignoni@fmed.edu.uy (L.B.); monicac@fmed.edu.uy (M.C.); 2Departamento Básico de Medicina, Facultad de Medicina, Universidad de la República, Montevideo 11600, Uruguay; nartagave@fmed.edu.uy; 3Departamento de Métodos Cuantitativos, Facultad de Medicina, Universidad de la República, Montevideo 11800, Uruguay; vcolistro@fmed.edu.uy; 4Departamento de Antropología Biológica, Facultad de Humanidades y Ciencias, Universidad de la República, Montevideo 11200, Uruguay; msans@fhuce.edu.uy; 5Departamento de Medicina Preventiva, Faculdade de Medicina, Universidade de São Paulo, São Paulo 01246-903, Brazil; cxbonilla@usp.br; 6Population Health Sciences, Bristol Medical School, University of Bristol, Bristol BS8 2BN, UK

**Keywords:** breast cancer, population genetics, Latin America

## Abstract

Among Latin American women, breast cancer incidences vary across populations. Uruguay and Argentina have the highest rates in South America, which are mainly attributed to strong, genetic European contributions. Most genetic variants associated with breast cancer were described in European populations. However, the vast majority of genetic contributors to breast cancer risk remain unknown. Here, we report the results of a candidate gene association study of sporadic breast cancer in 176 cases and 183 controls in the Uruguayan population. We analyzed 141 variants from 98 loci that have been associated with overall breast cancer risk in European populations. We found weak evidence for the association of risk variants rs294174 (*ESR1*), rs16886165 (*MAP3K1*), rs2214681 (*CNTNAP2*), rs4237855 (*VDR*), rs9594579 (*RANKL*), rs8183919 (*PTGIS*), rs2981582 (*FGFR2*), and rs1799950 (*BRCA1*) with sporadic breast cancer. These results provide useful insight into the genetic susceptibility to sporadic breast cancer in the Uruguayan population and support the use of genetic risk scores for individualized screening and prevention.

## 1. Introduction

Breast cancer is the most common cancer among women in populations of European descent, representing 25% of all cancers [1]. As in any complex disease, there are multiple genetic and environmental factors involved in its development. Risk factors, such as hormonal levels and lifestyle, have been reported as some of the most important [2]. However, genetic alterations underlying breast cancer have not been fully described, and only 15 to 30% of cases are heritable or due to genetic transmission. Susceptibility genes with high or moderate penetrance (i.e., *BRCA1, BRCA2, PALB2, ATM*, and *CHEK2*) have been identified in 30 to 40% of patients with a family history of breast cancer, but only in 5% of sporadic breast cancer cases. A large proportion of heritability still needs to be explained by low-penetrance variants and, therefore, a major effort has been underway to identify single-nucleotide polymorphism (SNPs) associated with breast cancer risk. These SNPs are not very informative by themselves, but together may account for almost half of the familial relative risk (FRR) [3]. A set of 170 SNPs strongly associated with breast cancer was described for familial cases, explaining up to 18% of the FRR in European populations [3]. Breast cancer heterogeneous incidences and mortality rates indicate a complex interaction between risk factors, one of the most relevant being a population’s geographic location [1]. The higher rates of incidence and mortality observed in populations of European descent suggest a population specific association [1,4]. These marked differences between European and non-European populations are expected to have an impact on the clinical presentation of breast cancer in admixed populations [5,6].

Latin American countries show great disparities in incidence and mortality rates across the continent. Uruguay, Argentina, and southern Brazil have incidence rates as high as southern Europeans (72, 72, and 70 cases per 100,000 women per year, respectively), while Bolivia has an incidence rate of 13 cases per 100,000 women per year [4]. This translates into almost 2000 annual cases of breast cancer for Uruguay. Uruguay also has the highest breast cancer mortality rate in Latin America [7]. The reasons for these high rates are not well understood. Environmental factors, such as increased consumption of red meat and fat, as well as a reduced intake of vegetables, have been identified as risk factors for breast cancer in Uruguay [8]. However, genetic factors associated with the European ancestral genetic contribution could also be playing an important role. In fact, countries with the highest breast cancer incidence in Latin America exhibit considerable European genetic contributions to their populations. For instance, urban admixed populations from Argentina, Brazil, or Uruguay have European genetic contribution ranging from 70 to 90%, whereas European ancestry in Bolivia amounts to less than 20% [9,10].

In Latin America, genetic analyses have been focused on *BRCA1* and *BRCA2*. Founder mutations that have been described in Chile, Brazil, Colombia, Uruguay, and Argentina can be explained by the complex demographic history of Latin American populations. The admixture and demographic dynamics promoted the drift to higher frequencies of rare variants [11]. Almost all Latin American countries show more mutations in *BRCA1* than in *BRCA2*, as observed in European countries, except from Uruguay, Costa Rica, Puerto Rico, and Cuba, with higher numbers of BRCA2 mutations [12]. 

The attempt to understand the genetic contribution to breast cancer in Latin America was centered in familial, hereditary, and early onset breast cancer, sometimes without making a distinction between patients regarding familial history or age at diagnosis [13]. Just a few studies of sporadic breast cancer have been conducted in Latin America, and they have focused on different candidate genes than our study [14,15,16,17].

In Uruguay, European ancestral proportions from autosomal markers show no substantial association with breast cancer, but a weak association of European mitochondrial haplogroup H with breast cancer risk was found in a previous study [18]. This represents a controversial issue; genetic ancestry seems to be a risk factor for breast cancer in Mexican and Hispanic women [19]. In other populations, from the Asia-Pacific region, and Europe, breast cancer risk is correlated with other factors, such as body mass index (BMI) [20,21]. The heterogeneous composition of the Latin American populations represents a challenge to understand the development of breast cancer. A higher number of specific-population mutations have been described for Uruguay compared to neighboring countries Argentina and Brazil [22,23,24].

However, given the strong European genetic contribution, the small population size, and quasi-null immigration since the 1950s, it is likely that the same candidate gene variants found amongst breast cancer cases in Europe could be increasing disease risk in Uruguay as well.

In this study, we analyzed a set of 144 candidate gene SNPs, previously associated with sporadic breast cancer in European patients, in the Uruguayan population. Common susceptibility loci are reported in genes whose rare variants explain a major fraction of hereditary breast cancer risk, which may, together, explain an appreciable fraction of the genetic variation in sporadic breast cancer.

## 2. Materials and Methods

### 2.1. Study Population

A case control study was carried out in public hospitals and private clinics across Uruguay between 2010 and 2012. Women were eligible if they were over 45 years old and had been diagnosed with sporadic breast cancer in the previous year. Controls were women of the same ages with normal mammograms and no familial history of breast or ovarian cancer, randomly selected from the same health centers as the patients. Participation involved completing an interviewer administered risk factor questionnaire and giving a blood sample. All patients provided written informed consent for participation in all aspects of the study. The details of the 416 women sampled, the recruitment strategy, participation, and data collection methods have been described previously [18]. The present study was conducted in accordance with the Declaration of Helsinki, and the protocol was approved by the Ethics Committee of Facultad de Medicina, Universidad de la República (071140-000439-06). 

### 2.2. DNA Extraction and Genotyping.

Genomic DNA was extracted from peripheral blood leukocytes using a FlexiGene^®^ DNA Kit (QIAGEN, Hilden, Germany) and stored at −20°C until analysis. SNP genotyping was undertaken by LGC-KBioscience Ltd. (Teddington, UK), who use their own form of competitive allele specific PCR system (KASPar^TM^), and also at the University of Chicago using Sequenom Mass Array technology (Sequenom Inc., San Diego, CA, USA) [25]. DNA extraction was successful for 189 cases and 211 controls. 

A panel of 141 SNPs in 98 candidate genes was genotyped in a sample of 176 patients and 183 controls, 13 patient samples, and 28 control samples failed during the process. The panel included 133 SNPs described by the Collaborative Oncological Gene-Environment Study (COGS) for breast cancer and 8 highly polymorphic SNPs from the HapMap project populations. SNPs with minor allele frequencies below 1% or genotyping rates lower than 90% were excluded. Samples with less than 90% genotyping rates were also excluded from further analysis. After filtering, 118 SNPs, 172 cases, and 160 controls remained for analysis (Appendix A). 

A set of 160 ancestry informative markers (AIMs) were genotyped along the autosomes and the X chromosome. The individual ancestral contribution from European, Native American, and African populations was estimated using STRUCTURE software, as described previously [18].

### 2.3. Statistical Analysis

Deviation from Hardy–Weinberg equilibrium (HWE) was ascertained for each SNP in cases and controls. The allelic association test was applied to individual SNPs, and multiple logistic regression adjusting for age, body mass index (BMI), and individual ancestry [18] effects was performed with PLINK [26]. Correction for multiple testing was carried out using the FDR method. Power estimation, considering an effect size of 0.26 (the highest from the COGS publication) and a confidence level of 95% was done in R [3,27].

### 2.4. Randomized Analysis

A randomized analysis (RDM) was implemented to account for confounder differences between cases and controls in a more efficient way. Therefore, a sub-sampling method without replacement was applied to create a set of sub-samples matched by each one of three variables (age, education, and socioeconomic status (SES)). A set of 1000 random sub-samples, each consisting of 100 cases and 108 controls, was generated. To check the homogeneity of each sub-sample, a chi-square test or t test was performed. If the sub-samples showed significant differences (*p* < 0.05) between patients and controls in age, education, or SES, they were discarded. This approach is similar to the one used by Tsao and Ling [28], except that in our study the distribution of the results of the genetic association analyses was explored within each sub-sample instead of creating a unique sub-sample without outliers. As sampling without replacement generates a smaller sample, those SNPs that passed the quality control in each sub-sample were reanalyzed. We expect to obtain a more accurate estimation than a method with replacement like bootstrapping. The sampling method and the posterior analyses were done with R software [27].

We performed a logistic regression analysis to examine the association of individual SNPs and haplotypes with breast cancer risk in each subsample, adjusted for BMI, ancestry, and age. The results were summarized by the frequency of significant p-values (significance level 0.05). If the number of significant *p*-values exceeded 50% the SNPs or haplotypes were kept for further analysis. SNP and haplotype-specific odds ratios (ORs) and corresponding 95% CIs were estimated.

### 2.5. Haplotype Analysis

To improve the association analysis, haplotype frequencies were calculated for the genes with two or more SNPs in significant linkage disequilibrium (LD). However, to understand the genetic structure of *BRCA1, BRCA2, ESR1*, and *VDR* genes in the Uruguayan population, a set of SNPs was chosen with a worldwide representation. SNPs selected for the analysis were in linkage disequilibrium in European populations; rs8176193, rs16942, and rs8176092 for *BRCA1*, rs206081, rs11571787, and rs4942505 for *BRCA2*, rs1514347 and rs6912184 for *ESR1*, and rs7975232, rs1544410, rs2239182, and rs2239179 for VDR. Data from the 1000 Genomes database was used to calculate haplotype frequencies in Africans (GWD–Gambians from the West Division and YRI–Yoruba from Nigeria), Asians (CHB–Chinese from Beijing and CHS–Chinese from the South), Europeans (CEU–Utah residents with Northern and Western European ancestry, IBS–Iberian from Spain, and TSI–Italians from Tuscany), and Latin Americans (CLM–Colombians from Medellin, MXL–Mexicans from Los Angeles, PEL–Peruvians from Lima, and PUR–Puerto Ricans).

### 2.6. Population Structure 

Population structure was analyzed in the Uruguayan sample using the candidate gene panel of SNPs to explore these particular genomic regions. Principal component analysis (PCA) was performed using R software [27]. Populations used were Africans (YRI), Asians (CHB, CHS, and JPT—Japanese in Tokyo), and Europeans (CEU, IBS, and TSI). Data from patients and controls was phased using Eagle v2.4.1 and imputed using the 1000 Genomes Phase 3 database with Minimac3 [29,30]. Genotyped SNPs were imputed in order to assess imputation accuracy. High-quality imputation was demonstrated with mean concordance greater than 0.994. Imputed SNPs with deviations from Hardy–Weinberg equilibrium were filtered out of downstream analyses.

## 3. Results

After quality control, data was available for 118 SNPs in 172 cases and 160 controls. All of the variants genotyped were found to be under HWE in controls after FDR correction (Appendix A).

The logistic regression results for the eight SNPs associated with breast cancer are listed in Table 1. Although none of them remained significant after FDR correction, the association of SNPs in *CNTNAP2* and *VDR* genes with the disease is supported by the randomized analysis. 

All, but rs1799950 of *BRCA1*, are risk variants with moderate effects ranging from 38–88% enhanced risk. *BRCA1* rs1799950 minor allele (G) showed a protective effect of ~50% decreased risk for the carriers of this variant. 

Since *ESR1* rs2941740 had a genotyping rate lower than 90%, it was excluded from the traditional analysis. Nevertheless, it stands out with the highest percentage of significant sub-samples in the randomized analysis (86% as a single SNP and up to 91% within haplotypes) and is, therefore, worthy of mention. 

Despite *BRCA2* SNPs not showing an association with breast cancer using the traditional approach, this gene appears associated in the haplotype analysis (Table 2). The haplotype frequencies for *ESR1, FGFR2, VDR*, and *BRCA2* genes are presented in Table 2.

We found a statistically significant association between haplotypes in these genes and breast cancer risk by both methods of analysis. Even though, after FDR correction, none of them remained significant with the traditional approach, they are strongly supported by the RDM analysis.

*BRCA2, VDR*, and *ESR1* genes showed two significant haplotypes, one that acts like a protective factor, and the other that acts like a risk factor. Both *BRCA2* haplotypes and the ESR1 protective haplotype revealed a strong support from RDM. The *FGFR2* gene exhibited a strong association (also with a strong support from RDM) with breast cancer only for the protective haplotype, which includes the major allele from rs2981582. This SNP was previously found as significantly associated with the disease, its minor allele increasing risk. 

To understand the genetic structure of *BRCA1, BRCA2, ESR1*, and *VDR* genes in the Uruguayan population, a set of SNPs was chosen with a worldwide representation. The frequency of the Uruguayan haplotypes was compared with that of parental populations, i.e., African, European, and Asian populations (as a surrogate for Native Americans), and other admixed Latin American populations (Appendix A). These haplotypes from the selected set of SNPs show no statistically significant differences between patients and controls. In general, haplotype frequencies in the Uruguayan population are very similar to those in Puerto Rico (already similar in *BRCA2* frequencies) [12].

Figure 1 illustrates the geographic distribution of *VDR* haplotypes (rs7975232|rs1544410|rs2239182|rs2239179). Some differences are remarkable. For example, the *VDR* ACCC haplotype frequencies in Uruguay and Colombia are close to each other, but are higher than those in European populations.

The PCA helps to understand the general genetic structure of the samples. When the Uruguayan samples are compared against the European, African, or Asian populations, they show a clear European pattern. However, the controls are more homogeneous, while the patients comprise a more heterogeneous group (Figure 2).

## 4. Discussion

Breast cancer is a common cancer in Latin American countries. Previous studies in Latin American populations focused mainly on hereditary breast cancer and *BRCA1*/*2* mutations [31]. This study was performed in a sample of women with sporadic breast cancer genotyped for 118 SNPs in 98 genes. Most of the SNPs (110) are present in the iCOGs microarray [32]. The results indicate that there is evidence for an association between polymorphisms in well-known candidate genes for breast cancer and the disease, however, with multiple testing stringent criteria the associations faded. 

To the best of our knowledge, this is the first study to explore these candidate genes in sporadic breast cancer patients in South America. Studies conducted in other Latin American countries either examined hereditary breast cancer or did not take into account family history at the time of selecting the patients. The few sporadic breast cancer studies available analyzed a small number of SNPs that are different from those included in the present study [13]. 

The admixture structure in Latin American populations represents an important factor to be considered in breast cancer studies. In the past decades, admixture mapping was developed as a powerful method of gene mapping [33]. This method takes advantage of the linkage disequilibrium generated during the admixture process in recently admixed populations. It relies on the idea that genes are involved in population-specific diseases (such as breast cancer in Europeans) and could be identified by ancestry informative markers. Even though this method has strong theoretical support, few studies were performed using Latin American samples, and even fewer involving breast cancer [34]. Although our study was conceived as a candidate gene study, at a population level, the candidate gene SNP frequencies showed differences between geographic populations, with the Uruguayan population closely resembling European populations. This confirms what was observed in a previous study where there was no increased risk of breast cancer associated with European ancestry [18], contrary to findings in other Latin American and Hispanic populations [35,36]. This is controversial under the hypothesis that breast cancer is a population-specific disease. When we consider other diseases, a clear excess of European genetic ancestry was detected in melanoma patients on chromosome 16 [37], which is expected for this type of cancer [38]. Moreover, other diseases, such as hemoglobinopathies or diabetes, showed an increased representation of non-European mutations, indicating the genetic contribution of Native American or African populations as expected [39,40,41]. 

In familial breast cancer, *BRCA1* and *BRCA2* have been the primary targets in Latin American studies, and one of their main achievements is the detection of new variants only present in those populations. Our experimental design did not allow us to identify new variants in the Uruguayan population. The *BRCA1* or *BRCA2* haplotype architecture is not different from that present in Europeans or other Latin Americans. However, the *BRCA1* rs1799950 G variant described as benign in hereditary breast cancer, was protective in our study, in agreement with other sporadic breast cancer studies in European populations [42]. In fact, the rs1799950 G allele is present mainly in Europe (6%) and in Latin American populations (5%). However, it is important to consider that most of the variants known today were discovered in populations of European descent. The large proportion of European ancestry along with the small sample size of this study are probably contributing to us failing to discover additional haplotypes that could explain risk in the Uruguayan population. 

The sub-sampling approach provided a different approximation to the analysis of the variants. Using this method, a random set of sub-samples was created by a variable selection procedure. The method reveals that *ESR1*, *FGFR2*, *VDR*, *CNTNAP2*, and *BRCA2* have a weak association with sporadic breast cancer in the Uruguayan population. The reduced sample size and the bias detected for SES decreased statistical power and confounded the traditional analysis. Nevertheless, the randomized approach suggests that genetic risk factors for sporadic breast cancer in Uruguay are similar to those in other populations of European ancestry [43]. Moreover, *BRCA2* seems to be a good candidate gene to explore in the future given the information available from previous studies in hereditary breast cancer in Uruguay. In these previous studies more mutations were described in *BRCA2* than in *BRCA1* with some of them being population-specific mutations [22,44]. A plausible explanation is that the Uruguayan population is the most aged in South America [45], and *BRCA2* mutations are slightly more common between older women with sporadic breast cancer [46]. Moreover, an association of common variants in genes, such as *FGFR2* (rs2981582) and *MAP3K1* (rs16886165), with *BRCA2* mutant carriers, has been described [47], but it is not present in our study (data not shown). In addition, rs2981582 in *FGFR2*, rs16886165 in *MAP3K1* and rs3803662 in *TOX3* are associated with breast cancer in populations of European ancestry [48], and the first two were also associated with breast cancer in Uruguay. In U.S. Hispanics [49], the association is weaker, a difference likely explained by the high proportion of Native American genetic contribution to southwest Hispanics [50,51]. Even if the relationship is not causative, these SNPs may be proxies for major candidate genes.

The subsampling method allowed us to reintroduce *ESR1* rs2941740 in the haplotype analysis and the haplotype rs2941740_G|rs1999805_T|rs827423_C showed a strong support. In a recent study, the authors identified the 6q25 region, where *ESR1* is located, as associated with breast cancer in U.S. Latinas, Mexican, and Colombian patients. The rs2941740 is one of the 48 SNPs showing robust evidence of association in GWAS. The 6q25 region was also previously described as associated with breast cancer and Native American ancestry in U.S. Latinas. In this region, breast cancer risk decreases with higher Native American genetic contribution, which suggests that a particular genomic structure related with breast cancer risk may exist [34]. Most of the variants described in the Hoffman et al. study are common in non-European populations and no direct relationship with breast cancer could be established. The *ESR1* gene, which encodes the estrogen receptor α, is an important candidate gene in the 6q25 region and risk allele carriers show abnormal gene expression [52]. The mechanism behind gene expression regulation by the risk alleles seems to be unclear. There are examples of the influence that the variability in the genomic context can cause in the chromatin status of a chromosomal region in cancer, or cardiovascular diseases [53]. However, perhaps the most paradigmatic example is the adult persistent expression of the lactase gene due to a single nucleotide change in Europeans [54]. 

Recently, we described the influence of epigenetic ancestry in sporadic breast cancer. DNA genome demethylation was positively correlated with African ancestry in breast cancer patients, but not in control individuals. The data suggests that the usual DNA hypomethylation in breast cancer patients is reinforced by the genome structure due to population-specific contributions [55]. Complex interaction between population-specific variants and DNA methylation status, some of them synergistic, could be influencing overall risk. How each variant modifies this risk in specific populations still needs to be elucidated.

Even though we found a weak association between European ancestry and breast cancer in Uruguay [18], there are risk genes detected in Europeans that appear to be risk genes in Uruguay too. The data suggest that European-specific variants are important, but also that the ethnic genetic contribution might be affecting through mechanisms that need further exploration in the admixed population of Uruguay. 

## 5. Limitations

In this study, a weak association with well-known candidate genes *BRCA1, BRCA2, ESR1, VDR*, and *FGFR2* was found; however, these associations did not persist after FDR correction. Given the small sample size, particularly the number of patients collected, even though it was representative of the Uruguayan population, was not enough to detect the expected differences. The *a posteriori* power estimation, considering an effect size of 0.26 (the highest from the iCOGs publication) and a confidence level of 95%, is lower than 70% [3]. We had no data on cancer staging or tumor grade, so additional studies are required to clarify if the influence of candidate genes affects tumor aggressiveness or outcome in patients with different ancestral components.

## 6. Conclusions

Given the current interest in identifying risk factors for complex diseases in populations other than Europeans, it is informative to analyze candidate genes in Latin American populations with a high contribution from European parental populations. Our findings suggest that known candidate genes like *FGFR2, ESR1, VDR*, or *BRCA2* could be associated with breast cancer in Uruguay and other admixed Latin American populations.

## Figures and Tables

**Figure 1 genes-11-01272-f001:**
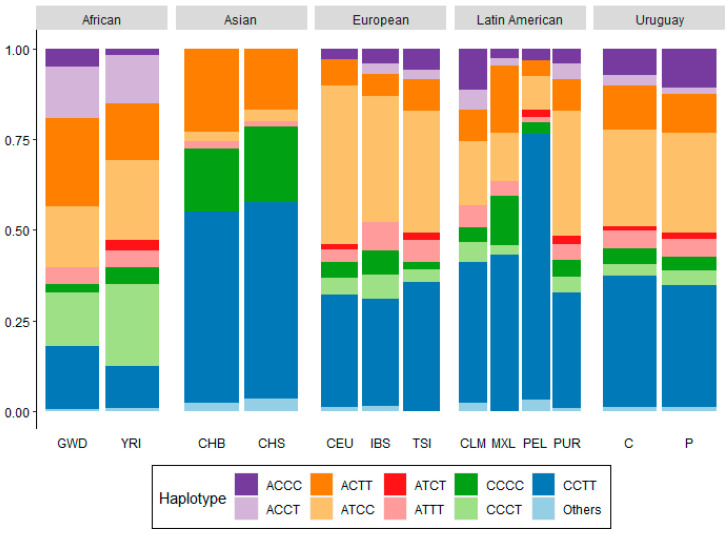
Haplotype frequencies of *VDR* SNPs for Uruguayan, Latin American, African, European, and Asian populations. P: patients, C: controls, CEU Utah Residents (CEPH), CHB: Chinese from Beijing, China, CHS: Chinese from Southern China, CLM: Colombians, GWD: Gambians, IBS: Iberian from Spain, MXL: Mexican from Los Angeles, USA, PEL: Peruvians, PUR: Puerto Ricans, TSI: Toscani from Italy, YRI: Yoruba from Nigeria.

**Figure 2 genes-11-01272-f002:**
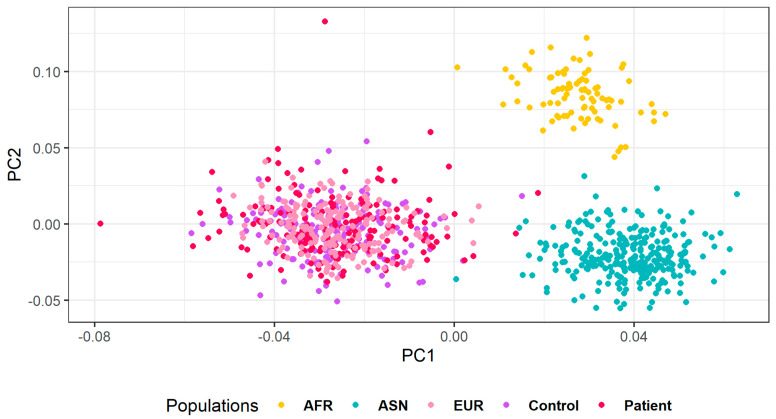
Principal component analysis (PCA) analysis for the SNPs in patients, controls, and African (AFR), Asian (ASN), and European (EUR) populations.

**Table 1 genes-11-01272-t001:** Analysis of individual SNPs using logistic regression adjusted for age, BMI, and individual European ancestry, and randomized sub-sampling.

GENE	Region	SNP	MAF_C_	MAF_P_	*p*-value	OR (95% CI)	RDM SES (%)
*MAP3K1*	5q11.2	rs16886165T>G	0.157	0.217	0.023	1.62 (1.07–2.47)	<50
*CNTNAP2*	7q35	rs2214681C>T	0.387	0.478	0.028	1.45 (1.04–2.02)	64
*FGFR2*	10q26.13	rs2981582C>T	0.396	0.469	0.024	1.52 (1.06–2.18)	<50
*VDR*	12q13.11	rs4237855G>A	0.425	0.526	0.036	1.38 (1.02–1.86)	50
*BRCA2*	13q13.1	rs144848A>C	0.249	0.311	0.080	1.38 (0.96–1.97)	<50
*RANKL*	13q14.11	rs9594759C>T	0.423	0.549	0.014	1.88 (1.14–3.12)	<50
*BRCA1*	17q21.31	rs1799950A>G	0.081	0.049	0.038	0.48 (0.24–0.96)	<50
*PTGIS*	20q13.13	rs8183919C>T	0.249	0.317	0.038	1.46 (1.02–2.10)	<50

MAF_C_: minor allele frequency in controls, MAF_P_: minor allele frequency in patients, *p*-value: adjusted *p*-value, OR: odds ratio, CI: confidence interval, RDM: percentage of randomized sub-samples with a significant *p*-value (<0.05). SES: socioeconomic status.

**Table 2 genes-11-01272-t002:** Haplotype association analysis of *ESR1, FGFR2, VDR, BRCA2* genes for the complete sample and randomized analysis.

GENE	SNPS	HAPLOTYPE	FP	FC	*p*-value	OR	RDM SES %
*ESR1 **	rs2941740|rs1999805|rs827423						
		GCC	0.043	0.029	0.212	2.06	<50
		GCT	0.050	0.048	0.619	1.28	<50
		GTC	0.212	0.133	0.005	2.05	91
		GTT	0.206	0.180	0.228	1.35	<50
		ACC	0.164	0.183	0.463	0.83	<50
		ACT	0.173	0.241	0.004	0.49	52
		ATC	0.050	0.072	0.162	0.53	<50
		ATT	0.102	0.113	0.598	0.84	<50
*FGFR2 **	rs1219648|rs2981582						
		AC	0.500	0.531	0.264	0.80	<50
		GC	0.012	0.058	0.023	0.22	99
		GT	0.488	0.412	0.098	1.40	<50
*VDR*	rs2238136|rs4237855						
		AA	0.031	0.017	0.278	1.67	<50
		AG	0.200	0.217	0.628	0.89	<50
		GA	0.495	0.406	0.031	1.36	55
		GG	0.274	0.360	0.026	0.69	60
*BRCA2*	rs144848|rs4987117						
		GC	0.360	0.239	0.007	1.75	89
		TC	0.640	0.761	0.008	0.58	86

FP: frequency in patients, FC: frequency in controls, P: *p*-value, RDM: percentage of randomized sub-samples with a significant *p*-value (< 0.05). SES: socioeconomic status. * Despite rs2941740 (*ESR1*) and rs1219648 (*FGFR2*) not being individually associated by the traditional method and therefore not present in Table 1, were included in the haplotype analysis because of the randomized sampling approach.

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
