# Peer review of "Genomic Diversity in Sporadic Breast Cancer in a Latin American Population"

_genes, 2020, doi:10.3390/genes11111272_

Round 1
Reviewer 1 Report
Overall, this study has high public health significance. The rationale for evaluating the risk variants in the Uruguayan population which has a strong ethnic similarity to the European ancestry has clinical relevance within the population for developing population specific risk scores. Additionally, it is interesting to understand how the similar genetic backgrounds of populations in different environments and heterogeneity with indigenous populations may impact cancer outcomes. Overall, I consider that the logic and approach for addressing the hypothesis is sound and that the evaluation of data is satisfactory. There are a few areas in which I recommend expanding the discussions to suit an international audience, provide clarity, and modifying the figures for visual improvement. My detailed recommendations are below.
- Lines 49-51, The authors present information regarding the differences in breast cancer incidence and mortality between the European and non-European populations within South America. I recommend that the authors explain this concept in more detail to inform the international readers that are not familiar with ethnic structures of these nations. It would be informative to understand the percentage of people with mixed, indigenous, or European ancestry in Argentina and Uruguay compared to Bolivia. This will further strengthen the understanding of the role of ancestry in genetic risk and its context in South America. Furthermore, it would be informative for the authors to briefly comment on the average age (or menopausal status) of breast cancer incidence, predominant molecular subtypes, stage at diagnosis in Uruguay and Bolivia. It would be informative to understand this information along with the genetic information.
- Line 159: The title for Table 1 seems to have a typographical error “ .4. Discussion”
- Table 1: A minor suggestion is to make in bold font the statistically significant OR column.
- Figure 1. I strongly recommend for the authors to include a higher resolution image and change the color scheme from gray scale to color or different patterns in the bars. It is very difficult to understand the lighter grey colors at the bottom.
- The concept of BRCA2 mutations being predominantly present in the Uruguayan population is an interesting point considering the cohort data also confirms this finding. In line 254-255, the authors mention that the Uruguayan population is mostly aged, and it could explain the predominance of BRCA2. According to the demographic characteristics of their cohort previously published from Bonilla et al (reference 18), although the mean age at diagnosis is statistically different among cases and control, it is within the same decade (56 vs 52) suggesting alternative or additional factors contributing to BRCA2 mutation predominance. It would be informative if the authors expanded on this point in more detail and provide additional discussion about the potential interaction between BRCA2 mutations and the common variants found. Is it known if there are worse (or better) health outcomes when BRCA2 interacts or is associated with the variants identified?
Author Response
Review 1
We thank the reviewer for their help in improving our manuscript. We consider that the background was improved. We tried to clarify the results (graphics and text) and conclusions for a better understanding of the relevance of this kind of studies in admixed populations.
Comment 1: Lines 49-51, The authors present information regarding the differences in breast cancer incidence and mortality between the European and non-European populations within South America. I recommend that the authors explain this concept in more detail to inform the international readers that are not familiar with ethnic structures of these nations. It would be informative to understand the percentage of people with mixed, indigenous, or European ancestry in Argentina and Uruguay compared to Bolivia. This will further strengthen the understanding of the role of ancestry in genetic risk and its context in South America. Furthermore, it would be informative for the authors to briefly comment on the average age (or menopausal status) of breast cancer incidence, predominant molecular subtypes, stage at diagnosis in Uruguay and Bolivia. It would be informative to understand this information along with the genetic information.
Response 1: We added more information about the genetic structure of Uruguay, Argentina, Brazil or Bolivia, and also we extended the discussion on this topic. Latin America represents a natural experiment to replicate previous results from other populations and to uncover new variants or mechanisms related to sporadic breast cancer, a complex disease. We added more information about the samples in the DNA extraction and genotyping section of Materials and Methods, however in the Discussion section we commented on the lack of patient information on the tumor stage for the Uruguayan sample.
Comment 2: Line 159: The title for Table 1 seems to have a typographical error “ .4. Discussion”
Response 2: We have corrected this typographical error.
Comment 3: Table 1: A minor suggestion is to make in bold font the statistically significant OR column.
Response 3: We have followed this suggestion in all tables.
Comment 4: Figure 1. I strongly recommend for the authors to include a higher resolution image and change the color scheme from grayscale to color or different patterns in the bars. It is very difficult to understand the lighter grey colors at the bottom.
Response 4: The errors were corrected and we included color images for Figure 1 and 2 with a higher resolution.
Comment 5: The concept of BRCA2 mutations being predominantly present in the Uruguayan population is an interesting point considering the cohort data also confirms this finding. In line 254-255, the authors mention that the Uruguayan population is mostly aged, and it could explain the predominance of BRCA2. According to the demographic characteristics of their cohort previously published from Bonilla et al (reference 18), although the mean age at diagnosis is statistically different among cases and control, it is within the same decade (56 vs 52) suggesting alternative or additional factors contributing to BRCA2 mutation predominance. It would be informative if the authors expanded on this point in more detail and provide additional discussion about the potential interaction between BRCA2 mutations and the common variants found. Is it known if there are worse (or better) health outcomes when BRCA2 interacts or is associated with the variants identified?
Response 5: We fully agree with the reviewer and have toned down the statement about aged women and BRCA2 mutations. We just wanted to point out that demography could be a factor to take into account in future studies. With respect to association analysis between BRCA2 mutations and common variations, the analysis was done but no significance associations were found and it is mentioned in the Discussion.
Reviewer 2 Report
The article “Genomic diversity in sporadic breast cancer in a Latin American population” by Brignoni et. al is within the scope of Genes. My biggest concern is that the article does not provide conclusive and significant results. The authors used Latin American population in their heading but just used data from one country Uruguay in Latin America and that too utilized a study which collected data a decade ago (2010-2012). The authors themselves state that number of patients used in this study is small and does not lead to any significant result.
Author Response
Review 2
We thank the reviewer for commenting on our manuscript. We consider that the background, the explanation of the methods and research design were improved. We tried to clarify the results (graphics and text) and conclusions for a better understanding of the relevance of this kind of studies in admixed populations
Comment: The article “Genomic diversity in sporadic breast cancer in a Latin American population” by Brignoni et. al is within the scope of Genes. My biggest concern is that the article does not provide conclusive and significant results. The authors used Latin American population in their heading but just used data from one country Uruguay in Latin America and that too utilized a study which collected data a decade ago (2010-2012). The authors themselves state that number of patients used in this study is small and does not lead to any significant result.
Response: We completely understand the reviewer's concern about the results of the study. We introduced more information about the Uruguayan population (small size, admixture and demographic characteristics) to justify why a candidate gene approach was done for this population (see Introduction and Discussion). Even though we couldn't reach significant statistical power we showed a trend for genes like BRCA1, 2, ESR1, FGFR2 and VDR using a different approach. As far as we know, this is one of the few studies in sporadic breast cancer performed in Latin America, and it is relevant for the region. We believed that the term “a Latin American population” avoids a narrow classification of the study.
Reviewer 3 Report
Breast cancer is the most common cancer diagnosed among Latinas and the leading cause of cancer-related death among this population. The presented studies are important for this reason, but in my opinion they are of a rather secondary nature. Dozens of papers on tumor genomics and pharmacogenomics in Latina patients with breast cancer have been published in the last 10 years. The main drawback of the research in this context is the low size of the study group (there are also inconsistencies in the methodology regarding the sample size), which translates into a low statistical significance of the results obtained (weak association with well-known candidate genes).
Nevertheless, the work is substantively well prepared, the methods and results are presented correctly (although poorly illustrated). There is no emphasis on the purpose of the study, the significance of the obtained results and the impact on diagnostics and clinical practice. Do these results have a potential impact on treatment regimens or are they only cognitive?
In addition, I find numerous editing, punctuation and linguistic errors at work.
Author Response
Review 3
We thank the reviewer for their help in improving our manuscript. We consider that the explanation of the methods and research design were improved. We tried to clarify the results (graphics and text) and conclusions for a better understanding of the relevance of this kind of studies in admixed populations
Comment 1: Breast cancer is the most common cancer diagnosed among Latinas and the leading cause of cancer-related death among this population. The presented studies are important for this reason, but in my opinion they are of a rather secondary nature. Dozens of papers on tumor genomics and pharmacogenomics in Latina patients with breast cancer have been published in the last 10 years. The main drawback of the research in this context is the low size of the study group (there are also inconsistencies in the methodology regarding the sample size), which translates into a low statistical significance of the results obtained (weak association with well-known candidate genes).
Nevertheless, the work is substantively well prepared, the methods and results are presented correctly (although poorly illustrated). There is no emphasis on the purpose of the study, the significance of the obtained results and the impact on diagnostics and clinical practice. Do these results have a potential impact on treatment regimens or are they only cognitive?
Response 1: We completely understand the reviewer's concern about the results of the study. As we explained before, we introduced more information about the Uruguayan population (small size, admixture and demographic characteristics) to justify why a candidate gene approach was done for this population (see Introduction and Discussion). With respect to the number of studies developed in Latino women, as far as we know germline variant studies in sporadic breast cancer are few, especially in the South region of the continent. We consider that the present results can help to understand the characteristics of the patients in the region.
We extended the discussion to overcome the lack of emphasis on the purpose of the study in the Discussion section.
Comment 2: In addition, I find numerous editing, punctuation and linguistic errors at work.
Response 2: We have made an extensive editing and english revision of the text.
Round 2
Reviewer 2 Report
The authors have addressed my comments. The paper is suitable for publication.
Reviewer 3 Report
Overview: This is an original article that presents some of the new information about genetic variants associated with breast cancer in Latin populations (local importance of research).
General Comments: The formatting and typos changes, improving the quality of figures made this revision much easier to read and are much appreciated. Overall, most concerns have been addressed. The occasional syntax error or odd word choice remains, but this can be corrected with careful copyediting. The biggest concern remaining is small group size and low statistical significance of the results. I appreciate the amount of work involved in improving the manuscript